# Pharmacokinetic Analysis of Enhancement-Constrained Acceleration (ECA) reconstruction-based high temporal resolution breast DCE-MRI

Zhen Ren[1,☯,*], Ty O. Easley[2,☯], Federico D. Pineda[3], Xiaodong Guo[1], Rina F. Barber[4], Gregory S. Karczmar[1]

1 Department of Radiology, The University of Chicago, Chicago, Illinois, United States of America,
2 McKelvey School of Engineering, Washington University in St. Louis, St. Louis, Missouri, United States of America, 3 Department of Radiology, The University of Pittsburgh, Pittsburgh, Pennsylvania, United States of America, 4 Department of Statistics, The University of Chicago, Chicago, Illinois, United States of America

☯ These authors contributed equally to this work.
* zren1@uhicago.edu

**Data Availability Statement:** All code is available at https://github.com/ZhenRen1/ECA-Demo. A minimal dataset are available at https://figshare.

## Abstract

The high spatial and temporal resolution of dynamic contrast-enhanced MRI (DCE-MRI) can improve the diagnostic accuracy of breast cancer screening in patients who have dense breasts or are at high risk of breast cancer. However, the spatiotemporal resolution of DCE-MRI is limited by technical issues in clinical practice. Our earlier work demonstrated the use of image reconstruction with enhancement-constrained acceleration (ECA) to increase temporal resolution. ECA exploits the correlation in k-space between successive image acquisitions. Because of this correlation, and due to the very sparse enhancement at early times after contrast media injection, we can reconstruct images from highly under-sampled k-space data. Our previous results showed that ECA reconstruction at 0.25 seconds per image (4 Hz) can estimate bolus arrival time (*BAT*) and initial enhancement slope (*iSlope*) more accurately than a standard inverse fast Fourier transform (IFFT) when k-space data is sampled following a Cartesian based sampling trajectory with adequate signal-to-noise ratio (SNR). In this follow-up study, we investigated the effect of different Cartesian based sampling trajectories, SNRs and acceleration rates on the performance of ECA reconstruction in estimating contrast media kinetics in lesions (*BAT*, *iSlope* and $K^{trans}$) and in arteries (Peak signal intensity of first pass, time to peak, and *BAT*). We further validated ECA reconstruction with a flow phantom experiment. Our results show that ECA reconstruction of k-space data acquired with 'Under-sampling with Repeated Advancing Phase' (UnWRAP) trajectories with an acceleration factor of 14, and temporal resolution of 0.5 s/image and high SNR (SNR ≥ 30 dB, noise standard deviation (std) < 3%) ensures minor errors (5% or 1 s error) in lesion kinetics. Medium SNR (SNR ≥ 20 dB, noise std ≤ 10%) was needed to accurately measure arterial enhancement kinetics. Our results also suggest that accelerated temporal resolution with ECA with 0.5 s/image is practical.

**Funding:** This study was supported by a grant from the National Institutes of Health under project number 5R01CA218700-04, which was awarded to G.S.K. The funders had no role in study design, data collection and analysis, decision to publish, or preparation of the manuscript.

**Competing interests:** The authors have declared that no competing interests exist.

# 1. Introduction

Breast dynamic contrast-enhanced magnetic resonance imaging (DCE-MRI) acquires a series of T1-weighted images before, during, and after the administration of a gadolinium-based contrast agent (GBCA). This provides dynamic information on the exchange of the GBCA between the vascular and interstitial compartments [1]. DCE-MRI is a valuable tool in oncology including early diagnosis [2], lesion classification [3], treatment planning [4] and treatment response assessment [5]. Analysis of DCE-MRI datasets is typically performed at different levels of complexity: (1) based on visualization of images, (2) based on semi-quantitative parameters derived from the signal enhancement curve as a function of time (e.g. arrival time of contrast media enhancement, enhancement rate or area under curve), and (3) based on pharmacokinetic modeling which allows quantification of tissue vascularization, perfusion, and capillary permeability.

Kinetic parameter accuracy relies heavily on an accurate measurement of tissue enhancement and GBCA concentration in feeding arteries, the latter is referred to as the 'arterial input function' (AIF). Direct measurement of the AIF is challenging and requires high temporal resolution to capture rapid changes in blood concentration of GBCA. A common solution for pharmacokinetic modelling is to use a general and well-defined AIF, which represents an average over many healthy patients [6]. However, this introduces errors into the parameter estimations as there is considerable variability in cardiac output even among healthy patients [7].

Simulation studies suggested that a temporal resolution of 1–1.5 second is necessary for accurate measurements of the AIF in each patient [7, 8]. In addition, high temporal resolution is needed to accurately detect the rapid uptake of GBCA that is used clinically to identify and characterize cancers. However, in clinical practice, high temporal resolution sampling is accompanied by loss of spatial resolution due to limited sampling rates. Therefore, clinical ultrafast breast DCE-MRI that produces full 3D bilateral breast scans achieves a temporal resolution ranging from 2–7 seconds depending on the field-of-view (FOV) and the number of slices [9].

MRI data exhibits a high degree of spatiotemporal correlation, resulting in redundant information in k-space. A basic strategy of modern acceleration strategy for DCE-MRI is to reduce the number of phase-encoding steps without loss of spatial resolution while maintaining adequate SNR, and to estimate the unmeasured k-space points with a reconstruction method. Many techniques have been introduced to accelerate temporal resolution of DCE-MRI. Popular methods include partial Fourier techniques [10], reduced FOV [11, 12], parallel imaging methods [13–16], view-sharing (VS) [17–21], compressed sensing [22, 23], and Kt-BLAST [24, 25]. However, their acceleration rates are limited due to the artifacts caused by missing high spatial frequency components [24, 25], the image processing [16] or non-uniform sampling [22]. In addition, parallel imaging reconstruction is heavily influenced by the estimation of coil sensitivities or their harmonic contributions [16]. The reconstruction error is more significant at higher acceleration factors [16].

To date, many regularization-based methods have been introduced to solve the ill-posed problem of reconstruction with highly under-sampled k-space data [26–31]. Most of them employ a total variation-based method for compressed sensing and required a non-Cartesian incoherent sampling [28–31]. Adluru *et. al.* [27] proposed a temporally constrained reconstruction based on an iterative $L_2$ norm regularization, but the method can only reach accelerations up to a factor of 5.

Our previous study proposed an enhancement-constrained acceleration (ECA) reconstruction from highly under-sampled k-space data with Cartesian sampling trajectory [32]. The strategy of ECA reconstruction is to recover missing k-space points by exploiting correlations,

based on two assumptions: (I) each k-space point contains some information about other k-space points and a properly selected subset of k-space points contains sufficient information to recover the rest of k-space; and (II) signal in each voxel enhances smoothly within short-time intervals. The temporal sampling frequency must be high enough to meet the smoothness criterion, but low enough to provide adequate signal-to-noise ratio. To obtain smooth enhancement in both arteries and suspicious lesions, we chose a target temporal resolution of 0.25 seconds in the previous study, based on published estimates of the optimal temporal resolution for breast DCE-MRI [6–8, 33]. Our previous study demonstrated that sparse, uniform k-space samples with adequate SNR can be used to reconstruct breast DCE-MRI with high fidelity. However, the performance of ECA reconstruction for measuring the shape of the AIF and lesion pharmacokinetics was not addressed in our earlier work. In addition, the impact of noise and acceleration factor on ECA reconstruction was not evaluated. Therefore, in this follow-up study, we focus on the performance of ECA reconstruction in accurately recovering the AIF and semi-quantitative kinetic features in lesions, as well as pharmacokinetic modelling. We also investigate the robustness of ECA reconstruction at different noise levels and acceleration factors with three Cartesian based sampling trajectories. In addition, we validate the ECA reconstruction using a flow phantom.

## 2. Methodology

### 2.1. Simulation phantoms

To provide quantitative ground truths for validating the reconstruction method, five *in silica* breast phantoms, representing a range of pathologies (Fig 1), were created from real clinical data; paired ultrafast and high-spatial resolution DCE-MRI datasets. Table 1 lists acquisition parameters for original patient MR scans.

The phantoms were created based on a set of parameters that defined the concentration of contrast media in each voxel as a function of time. This was converted from the signal intensity enhancement over time measured in each voxel from the original ultrafast DCE-MRI dataset

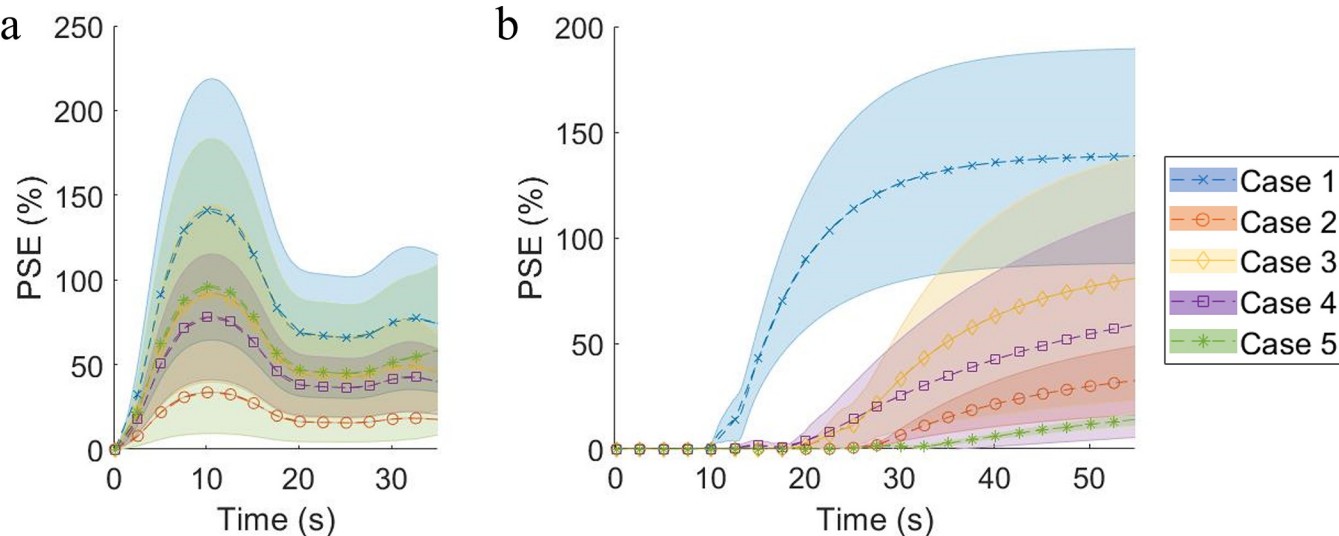

**Fig 1.** Percent signal enhancement (PSE) curve versus time in (a) vessel voxels and (b) lesion voxels by cases. Markers and dashed lines: mean of PSE; Area: standard deviation of PSE. PSE in vessel voxels plotted in (a) were measured relative to bolus arrival time (*BAT*). Case 1: Invasive ductal carcinoma (IDC), Grade III; Case 2: ductal carcinoma in situ (DCIS); Case 3: Invasive lobular carcinoma, Grade II; Case 4: IDC Grade III; Case 5: No abnormal enhancement (control case).

**Table 1. MRI parameters of the source patient images.**

|  | Standard | Ultrafast |
|---|---|---|
| TR/TE | 4.8/2.4 | 3.2/1.6 |
| Acquisition Voxel Size (mm$^3$) | $0.8 \times 0.8 \times 0.8$ | $0.8 \times 0.8 \times 2$ |
| Temporal Resolution Range (s/image) | 60–70 | 2.8–7 |
| Flip Angle (˚) | 10 | 10 |
| Field of View | 300–330 mm (X, Y)<br>160–200 mm (Z) | 300–330 mm (X, Y)<br>160–200 mm (Z) |
| Number of Slices | 200–250 | 80–107 |

acquired *in vivo*. Each phantom consisted of three categories of signal sources: blood vessel, lesion, and background, and each was modeled with a distinct function that computed the time-evolution of concentration of contrast media and then generated images of signal intensity enhancement during the scan at 50 msec intervals. The functions were used to fit the original data with high precision. As a result, we were able to evaluate the accuracy of measurement of kinetic features (e.g. BAT, AIF, and $K^{trans}$) at high temporal resolution from the ECA reconstruction using various k-space sampling methods, even though these features were measured at lower temporal resolution in the acquired data. Development of breast phantoms can be divided into three steps (Fig 2).

(1) Segmentation: blood vessels and lesions were segmented from patient ultrafast and standard high-spatial resolution DCE-MR images, which both underwent a non-rigid registration for motion correction [34]. 3D vasculature was segmented from subtracted high-spatial resolution images by a Hessian filtering process [35]. Lesions were segmented manually.

(2) Creating maps of kinetic parameters: ultrafast image sets were used to generate pharmacokinetic maps. Signal intensity of each lesion voxel was converted into the concentration of contrast agent over time [36]:

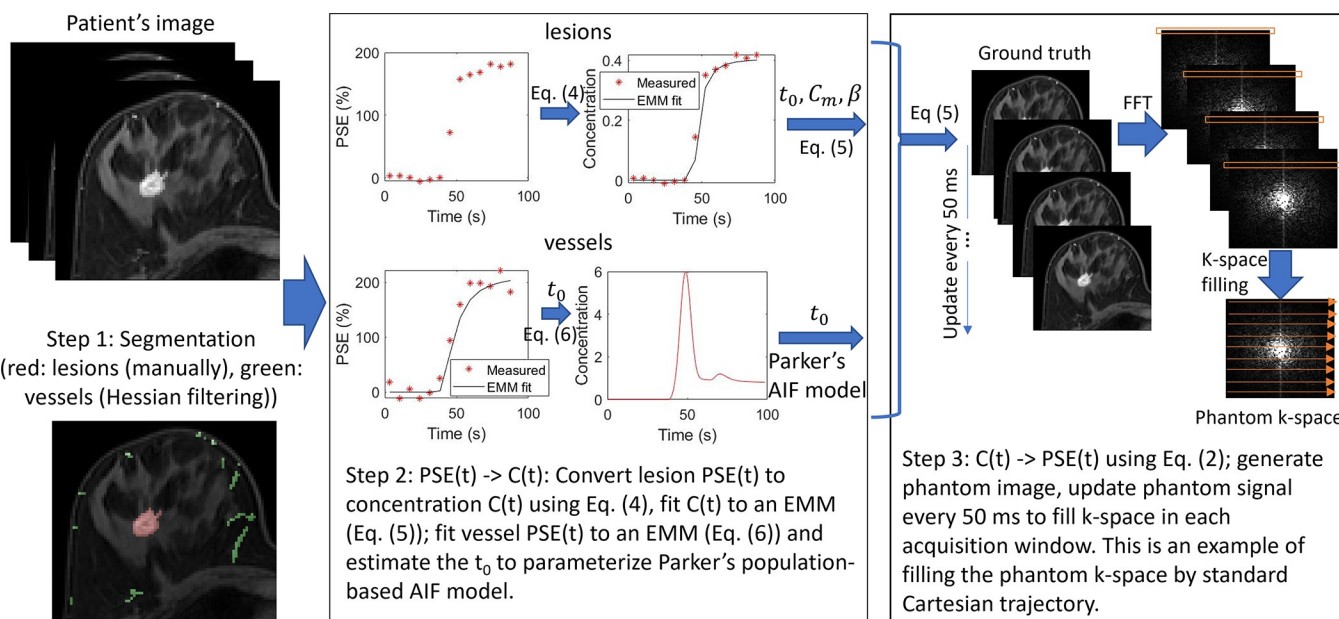

**Fig 2. Pipeline of the phantom development from patient images.**

$$C(t) = \frac{1}{r_1} \cdot \left( \frac{1}{T_1(t)} - \frac{1}{T_{10}} \right) \tag{1}$$

where $r_1$ is the relaxivity, $r_1 = 3.4$ mMol$^{-1}$s$^{-1}$ was used in all simulations [37]; $T_{10}$ (lesion) = 1.44 s [38]and $T_{10}$ (blood) = 1.66 s [39]; $T_1(t)$ is derived using the spoiled gradient-echo signal model:

$$S(t) = K \sin(FA) \times \frac{\left(1 - e^{-\frac{TR}{T_1(t)}}\right)}{1 - \cos(FA)e^{-\frac{TR}{T_1(t)}}} \tag{2}$$

where $K$ is a scaling constant which ensures the pre-contrast and post-contrast maximum signal intensity of each AIF and lesion voxel are consistent with the real images; FA = 10˚, TR = 3.2 ms.

$$T_1(t) = \frac{-TR}{\ln\left(A - \frac{S(t)}{S(0)}\right) - \ln\left(A - \frac{S(t)}{S(0)} \cdot \cos(FA)\right)} \tag{3}$$

where $A = \frac{1 - \cos(FA) \cdot e^{-\frac{TR}{T_{10}}}}{1 - e^{-\frac{TR}{T_{10}}}}$, S(t) and S(0) are the signal intensity across time t and pre-contrast signal intensity, respectively. We can then write Eq (1) as

$$C(t) = \frac{1}{r_1 \cdot TR} \cdot \left( \ln\left( \frac{A - \frac{S(t)}{S(0)}}{A - \frac{S(t)}{S(0)} \cdot \cos(FA)} \right) - \frac{1}{T_{10}} \right) \tag{4}$$

An empirical model used a truncated exponential function to fit the $C(t)$ in lesion during initial enhancement phase (uptake only):

$$C(t) = (t \geq t_0) \cdot C_m \cdot \left(1 - e^{-\beta(t-t_0)}\right) \tag{5}$$

where $t_0$ is the $BAT$ in lesion voxels (s), $C_m$ is the upper limit of tracer concentration (mmol), and $\beta$ is the uptake rate (s$^{-1}$).

Due to the use of full dose contrast agent, the AIF could not be accurately measured directly from arteries by ultrafast DCE-MRI[1, 40]. Instead, we used Parker's population-based AIF model [6] to generate the concentration change in vessel voxels. Percent signal enhancement $\left( PSE(t) = \frac{S(t) - S(0)}{S(0)} \right)$ of the first peak was fit into an empirical exponential function model similar to Eq (5)

$$PSE(t) = (t \geq t_0) \cdot A \cdot \left(1 - e^{-\alpha(t-t_0)}\right) \tag{6}$$

where $A$ is the upper limit of signal intensity, and $\alpha$ is the uptake rate (s$^{-1}$). $t_0$ calculated from Eq (6) was used to parameterize Parker's model [6] for each vessel voxel,

$$C_b(t) = (t > t_0) \cdot \sum_{n=1}^{2} \frac{A_n}{\sigma_n} e^{\left( -\frac{(t-t_0-T_n)^2}{2\sigma_n^2} \right)} + \frac{\gamma e^{-\mu(t-t_0)}}{1 + e^{-s(t-t_0-\tau)}} \tag{7}$$

where $A_n$, $T_n$ and $\sigma_n$ are the scaling constants, centers, and widths of the n$^{th}$ Gaussian; $\gamma$ and $\mu$ are the amplitude and decay constant of the exponential; $s$ and $\tau$ are the width and center of the sigmoid, respectively. The parameter values of the Parker's population AIF model are listed in Table 2.

**Table 2. Parameter values for Parker's AIF model [6].**

| Parameter | $A_1$ | $A_2$ | $T_1$ | $T_2$ | $\sigma_1$ | $\sigma_2$ | $\gamma$ | $\mu$ | $s$ | $\tau$ |
|---|---|---|---|---|---|---|---|---|---|---|
| Value | 0.809 | 0.330 | 0.17046 | 0.365 | 0.0563 | 0.132 | 1.050 | 0.1685 | 38.078 | 0.483 |
| Units | mmol·min | mmol·min | min | min | min | min | mmol | min$^{-1}$ | min$^{-1}$ | min |

In summary, kinetic maps of $t_0$, $C_m$, $\beta$ were used to parameterize the three-parameter empirically derived contrast enhancement model Eq (5) in lesion voxels, and a kinetic map of $t_0$ was used and combined with parameters in Parker's AIF model for vessel voxels. For background voxels, the $t_0$, $C_m$ and $\beta$ were set to 0.

(3) Generating phantom k-space: For each phantom, ground truth MR images were computed by converting concentration to signal by Eq (2), where $\frac{1}{T_1(t)} = \frac{1}{T_{10}} + r_1 C(t)$, which was derived from Eq (1).

To simulate signal evolution during the acquisition time, phantom signal was updated every 50 ms. This simulation was used to update k-space in each acquisition window. Fig 1 shows the mean and standard deviation of the percent signal enhancement ($PSE(t)$) curve versus time in arteries and in lesions by case, where the PSE curves in vessel voxels were measured relative to $BAT$ in the voxel.

For each phantom, the nominal temporal resolution (the time required for complete cartesian sampling for each image) was 7 seconds, and the full acquisition time was 56 seconds for the entire DCE series.

## 2.2. K-space sampling trajectories

We compared three Cartesian-based k-space sampling trajectories: standard Cartesian and two under-sampling protocols based on Repeated Advancing Phase (UnWRAP). In standard Cartesian sampling, the k-space was filled from edge to edge via a row-by-row trajectory.

In UnWRAP trajectories, the time of acquisition for each phase-encoding line depends on the acceleration rate. For a N-fold acceleration, in UnWRAP1, we split all phase-encoding lines of k-space into N sections, each separated into N sheaves, so that each sheaf contained 1/$N^2$ of total phase-encoding lines in k-space. We acquired the first sheaf of phase-encoding lines in the first section, then moved to the first sheaf in the second section. After acquiring the first sheaf in the last section, we moved to the acquisition for the second sheaf for all sections. The acquisition continued until the entire k-space was filled (Fig 3C and 3D). Within each sheaf, k-space was sampled in a Cartesian manner. In UnWRAP2, we first sampled every Nth phase-encode line from the first line, and then we sampled every Nth phase-encode line from the second line and looped until all phase-encoding lines were acquired (Fig 3E and 3F). Within each loop, k-space was sampled in a Cartesian manner. This is similar to the trajectory used in Kt-BLAST but without denser sampling in the central k-space region. Fig 3 shows the patterns of the three trajectories when acceleration factor = 5, where frequency-encoding direction is omitted, and each frequency-encoding line is represented as a dot for simplicity.

## 2.3. Image reconstruction

**2.3.1. Enhancement-constrained acceleration (ECA) reconstruction.** ECA reconstruction assumes that the enhancement is smooth on the timescale of the reconstruction's temporal resolution. Sharp changes in signal between consecutive reconstructed time points are penalized. We constrain the new images to match the k-space data sampled over each reconstructed time interval. Intuitively speaking, ECA searches for the smoothest set of

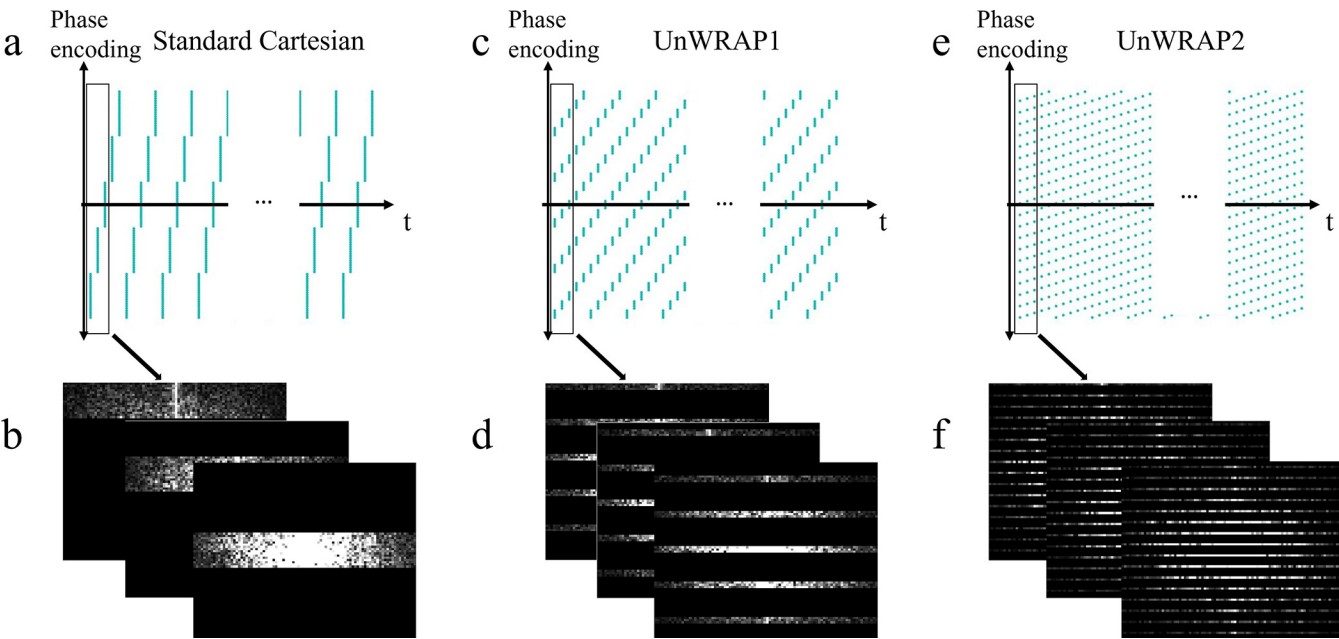

**Fig 3. Illustration of sampling trajectories.** Sampling pattern (upper) and k-space sampled for the first three time points (bottom) with a 5-fold acceleration: (a)(b) standard Cartesian, (c)(d) UnWRAP1, and (e)(f) UnWRAP2.

enhancement curves consistent with the highly under-sampled k-space data measured during each of the reconstructed time intervals. A positive semidefinite smoothness penalty matrix is used to penalize the discretized second derivative in temporal dimension [32]. The ill-posed inverse problem is then solved by conjugate gradient method. The details of the ECA reconstruction are provided in [32].

**2.3.2. Inverse Fast Fourier Transform (IFFT) reconstruction.** To benchmark the ECA reconstruction, the IFFT which is one of the standard MR Image reconstruction methods, was applied to reconstruct MR images from fully sampled k-space data (temporal resolution of 7 s/image).

## 2.4. Flow phantom experiment

To further evaluate the performance of the ECA reconstruction, i*n-vitro* flow measurements were acquired on a flow-kinetic phantom (Fig 4). This phantom was designed to produce realistic input and tissue contrast uptake and washout curves by adjusting downstream flow rates between two outputs [41]. The phantom consists of a coiled tube within a water-filled plastic container, with water pumped through the tube by a peristaltic flow pump at a constant flow rate of 300 ml/min from outside the scan room. The flow phantom was imaged in a Philips Ingenia 3.0-T MRI scanner equipped with a 32-channel head coil. The DCE-MRI protocol employed a standard T1-weighted GRE sequence to obtain 30 serial 2D images of the central axial plane of the flow phantom, with a temporal resolution of 4.9 s/image. Ground truth images were acquired with a temporal resolution of 0.56 s/image. Table 3 listed the DCE-MRI parameters for acquiring the ground truth images and the measurement images. A multi-shot linear Cartesian sampling trajectory was selected, with TFE factor = 14 and TFE shot = 100. The linear sampling profile was rearranged according to the TFE shot to provide an UNWRAP1 trajectory with an acceleration factor = 10. Contrast bolus of diluted 1 ml Multi-Hance (Bracco, NJ, diluted 20:1) was delivered and followed by a saline flush of 10 ml, both at 0.2 ml/s.

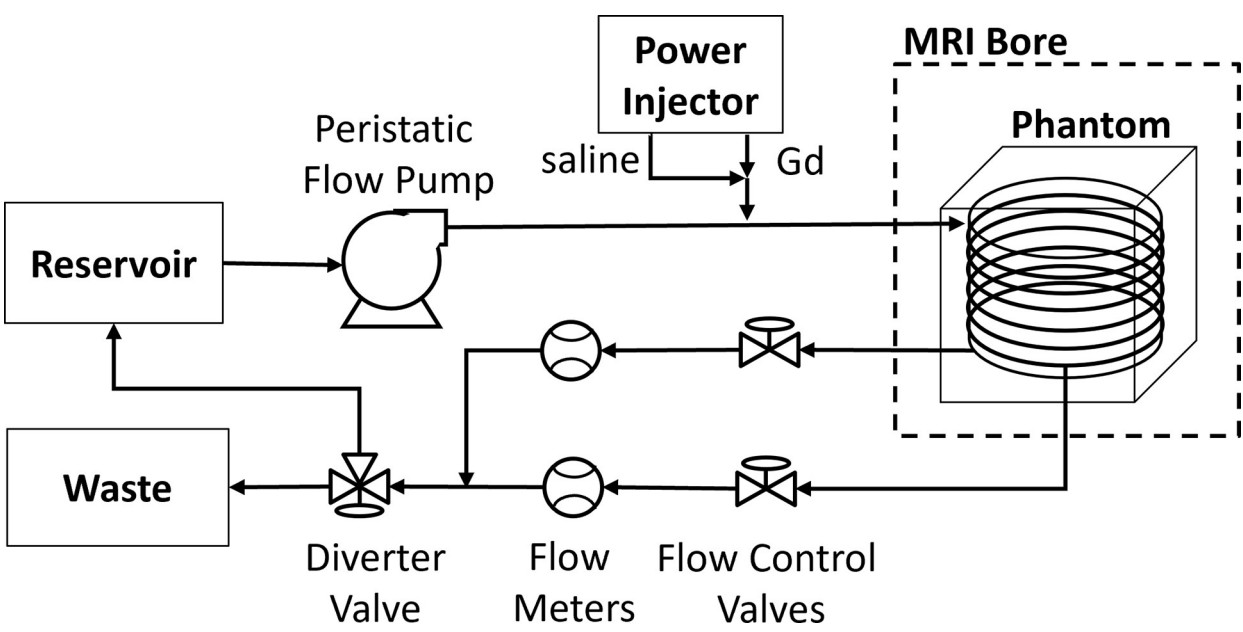

**Fig 4. Schematic representation of the flow phantom system, consisting of pump, contrast injection (power injector), coiled input, branching into two outputs with relative flow controlled by pinch valves and measured by digital meters.**

## 2.5. Data analysis

To compare the performance of ECA reconstruction to the standard IFFT, an interpolation method was used to fit the signal enhancement curves at each voxel and to interpolate the curves from ECA images and IFFT images so that the same measurements are simulated at 50 ms intervals. In this study, we employed a modified Akima piecewise cubic Hermite interpolation (Makima) method to interpolate the enhancement curve in the time series of each voxel within ROIs. Makima is a non-smoothing spline based on a piecewise function composed of a set of polynomials [42]. It gives a good fit to the curve where rapid change occurs between flat regions. All simulation kinetic features were measured from Makima interpolated enhancement curves extracted from ECA and IFFT images.

Signal enhancement curves of a vessel and a lesion voxel from Case 1 are shown in Fig 5, where the curves from images reconstructed by ECA and IFFT, and by different sampling trajectories are compared to the ground truth. For each vessel and lesion voxel, we fit the interpolated $PSE(t)$ to the EMM using Eq (6), obtaining primary parameters $BAT$, $A$ and $\alpha$ Then, the secondary parameter, initial enhancement slope, was calculated as: $iSlope = A \cdot \alpha$ (s$^{-1}$). Note that for the vessel voxels, we used Eq (6) to fit the enhancement curve to the AIF first pass to estimate $BAT$.

**Table 3. DCE-MRI parameters of the flow phantom experiment.**

|  | Ground Truth | UnWRAP1 Measurement |
|---|---|---|
| **TR/TE** | 8.3/4.1 | 8.3/4.1 |
| **Acquisition Pixel Size (mm$^2$)** | 2.5 × 2.5 | 0.47× 0.57 |
| **Temporal Resolution (s/image)** | 0.56 | 4.9 |
| **Flip Angle (°)** | 10 | 10 |
| **Field of View (mm$^2$)** | 230 × 163 | 230 × 228 |
| **TFE factor** | 1 | 14 |
| **Slice Thickness (mm)** | 5 | 5 |

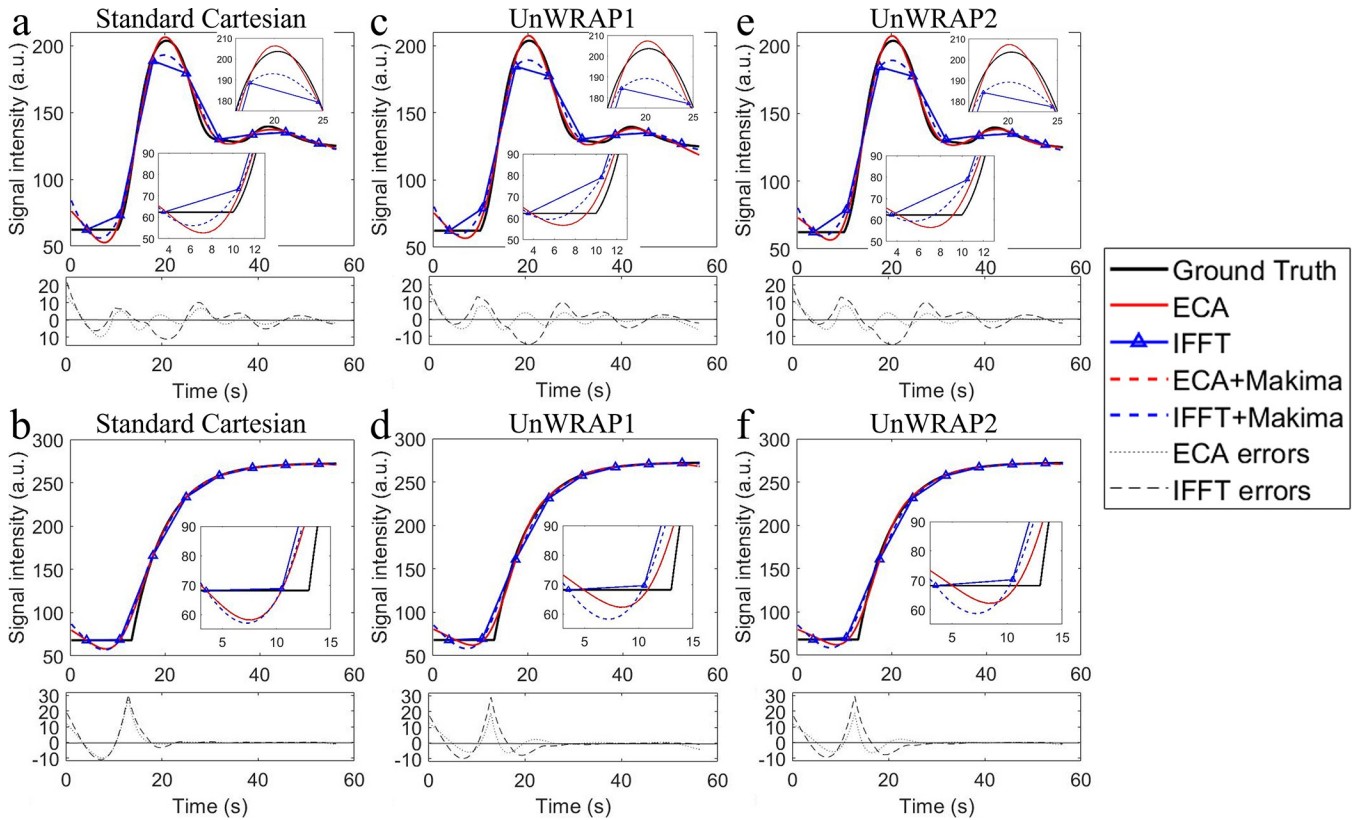

**Fig 5.** Signal enhancement curves of a vessel voxel (upper) and a lesion voxel (bottom) from ground truth and images reconstructed by ECA and standard inverse fast Fourier transform (IFFT) with k-space sampled by (a) (b) standard Cartesian, (c) (d) UnWRAP1, and (e) (f) UnWRAP2. Images were acquired noise-free phantom. Temporal resolution of IFFT is 7 seconds, ECA is 0.5 seconds (acceleration factor = 14). Errors = measured (interpolated)–ground truth.

For lesion voxels, we measured $K^{trans}$, the volume transfer constant between blood plasma and extravascular extracellular space. We calculated $K^{trans}$ from a signal intensity form of the Tofts model, which was introduced in an earlier study [43]:

$$PSE(t) = \frac{S_b(0)}{S_t(0)(1 - Hct)} K^{trans} \int_0^t PSE_b(\tau) \exp\left(-\frac{(t - \tau)K^{trans}}{v_e}\right) d\tau \qquad (8)$$

where $S_b(0)$ and $S_t(0)$ are the baseline signal intensity in blood vessel and in lesion, respectively. $PSE_b(t)$ and $PSE_t(t)$ are the percent signal enhancement in blood vessel and in lesion, $Hct$ is the hematocrit ($Hct = 0.42$) [6], $v_e$ is the volume of EES per unit volume of tissue $v_e \in [0, 1]$. Note that for each case, $S_b(0)$ and $PSE_b(t)$ were measured from a single voxel at the feeding artery.

To evaluate the performance of ECA reconstruction for vessel voxels, in addition to the $BAT$, we studied two parameters that are directly measured from the interpolated $PSE(t)$ curves:

- AIF peak signal intensity of the first pass in arteries ($SI_{peak}$) is the maximum signal enhancement value across the time of measurement: $SI_{peak} = \max(PSE)$.

- Time at the first peak: $T_{peak} = t$ when $PSE(t) = SI_{peak}$.

Moreover, we measured *Error ratio* to compare the performance of ECA reconstruction to that of the standard IFFT,

$$Error\ ratio = \frac{P_{ECA} - P_{GT}}{P_{IFFT} - P_{GT}} \tag{9}$$

where $P_{ECA}$ and $P_{ifft}$ refer to kinetic parameter measured from images reconstructed by ECA reconstruction method and by IFFT, respectively, $P_{GT}$ is the ground truth value of the kinetic parameter.

To investigate the performance of ECA reconstructions with noise, white Gaussian noise was added to k-space. Three signal-to-noise-ratios (SNRs) - 30 dB, 20 dB and 10 dB–were investigated in this study. As a reference, the SNR of clinical ultrafast DCE-MRI ranges from 20 to 30 dB when measured as the averaged post-contrast signal intensity in the time series divided by the standard deviation of the baseline signal. Tumor and vessel regions of interest (ROI) have a higher SNR (>30 dB) due to the greater signal enhancement.

To investigate the impact of acceleration factor on estimation of kinetic features, four acceleration factors were investigated: 5, 10, 14, and 20, which accelerated the nominal temporal resolution of 7 seconds to 1.4, 0.7, 0.5, and 0.35 seconds, respectively.

In the flow phantom experiment, 10-fold ECA reconstruction was used to recover the signal enhancement curve in the tube ROIs. To quantify the performance of the ECA reconstruction, we compared the $SI_{peak}$ and delay time ($T_D$) measured from the ECA and IFFT images to the ground truth. $T_D$ was calculated as the time lag between the relative enhancement signal of ROI1 (inflow) and the other ROIs by finding the largest absolute value of the normalized cross-correlation of the two signals. For comparing measured signal to the ground truth at each time point, Makima interpolation was used for ECA and IFFT images to interpolate the signal at the same time point as the ground truth acquisition. Absolute error was calculated by taking the absolute value of the difference between the parameters measured from reconstructed images and from ground truth images. SNR of the flow phantom image was calculated by mean of the enhanced signal in the tube ROIs divided by standard deviation (std) of the background area.

## 2.6. Statistical analysis

The 'Bootstrap' method was used for calculating confidence intervals. The Wilcoxon rank sum test was used to compare kinetic parameters between different reconstruction methods and different sampling trajectories. A value of $p < 0.05$ was considered statistically significant.

## 3. Results

### 3.1. ECA vs IFFT with different sampling trajectories

ECA reconstructions recovered signal enhancement curves more accurately than standard IFFT, as shown by a smaller error variance across the 'scan' time for ECA (Fig 5), particularly when rapid signal change occurred, e.g. during the first pass of the AIF. This is critical for precise measurements of kinetic features. With any sampling trajectories, ECA reconstruction estimated each lesion and AIF kinetic parameter more accurately than IFFT ($p < 0.0001$). To demonstrate this quantitatively, Table 4 shows the median value and the 95% confidence intervals (CIs) of the *Error ratio* of ECA and IFFT by kinetic features and by sampling trajectories. All the kinetic parameters were estimated from noise-free images with a 14-fold acceleration factor. All the median values of the *Error ratio* were below 1, referring to a lower error measured by ECA than IFFT. *Error ratio* values varied by parameter and by sampling trajectories.

**Table 4. Median value and the 95% confidence interval (CI) of voxel based ECA vs IFFT *Error ratio*, noise free, acceleration factor = 14.**

| Parameters | | UnWRAP1 | | UnWRAP2 | | Standard Cartesian | |
|---|---|---|---|---|---|---|---|
| | | Median | 95% CI | Median | 95% CI | Median | 95% CI |
| Lesions | $iSlope$ | **0.215** | **[0.209, 0.222]** | 0.233 | [0.230, 0.236] | 0.810 | [0.798, 0.820] |
| | $BAT$ | 0.178 | [0.172, 0.184] | **0.167** | **[0.163, 0.172]** | 0.775 | [0.764, 0.786] |
| | $K^{trans}$ | 0.050 | [0.047, 0.052] | **0.038** | **[0.037, 0.040]** | 0.643 | [0.640 0.646] |
| AIF | $SI_{peak}$ | **0.508** | **[0.497, 0.513]** | 0.522 | [0.514, 0.530] | 0.526 | [0.491, 0.558] |
| | $T_{peak}$ | 0.227 | [0.227, 0.235] | **0.200** | **[0.200, 0,200]** | 0.238 | [0.213, 0.250] |
| | $BAT$ | 0.166 | [0.165, 0.169] | **0.160** | **[0.158, 0.162]** | 0.486 | [0.475, 0.500] |

**Bold highlights the lowest *Error ratio* among the three trajectories**

*BAT*: bolus arrival time

*iSlope*: Initial enhancement slope

*$SI_{peak}$*: Peak signal intensity of the first AIF pass

*$T_{peak}$*: time at peak signal of the first AIF pass

Of the three sampling trajectories, UnWRAP1 and UnWRAP2 with ECA reconstruction measured vessel and lesion kinetics much more precisely than with IFFT (median *Error ratio* range = 0.038–0.522). The difference between ECA and IFFT was smaller with standard Cartesian sampling (median *Error ratio* range = 0.238–0.81). Among all vessel and lesion kinetic features, ECA with UnWRAP2 measured $K^{trans}$ much more accurately than IFFT with UnWRAP2, with median *Error ratio* = 0.038 (95% CI (0.037, 0.04)).

Fig 6 compares the performance of ECA reconstruction with the three sampling trajectories. Comparing the two UnWRAP trajectories, UnWRAP1 and UnWRAP2 were not significantly different when estimating lesion *iSlope* (p = 0.99) and lesion *BAT* (p = 0.63). ECA reconstruction with UnWRAP2 trajectory measured lesion $K_{trans}$, AIF $T_{peak}$, and AIF *BAT* more accurately than ECA with UnWRAP1 (p<0.0001), while ECA with UnWRAP1 estimated AIF $SI_{peak}$ more accurately than ECA with UnWRAP2 (p <0.0001). The performance of ECA with standard Cartesian sampling was significantly different from ECA with UnWRAP trajectories (p<0.05). For all lesion and arterial kinetics except AIF $SI_{peak}$, ECA with the two UnWRAP trajectories provided more accurate measurement than ECA with standard Cartesian sampling, showing a lower median error and smaller error variance. However, ECA with standard Cartesian sampling showed higher accuracy in estimating AIF $SI_{peak}$ (p < 0.0001) than UnWRAP trajectories, although the error variance was much greater.

### 3.2. ECA performance with noise

Fig 7 shows the absolute (percent) error in lesion kinetics measured from ECA and IFFT images, by different sampling trajectories with varying SNR. ECA with UnWRAP trajectories showed lower median absolute (percent) errors than ECA with standard Cartesian or IFFT with any trajectories. To ensure a minor error—error < 5% or < 1 s—high SNR (SNR ≥ 30 dB, noise std < 3%) was needed for sampling with UnWRAP trajectories.

Fig 8 shows the absolute (percent) error in AIF kinetics measured from ECA and IFFT images, for different sampling trajectories and with different SNRs. ECA reconstruction with k-space sampled with a standard Cartesian trajectory estimated AIF $SI_{peak}$ more accurately than either of the UnWRAP sampling trajectories. This is shown by smaller median absolute percent errors estimated by standard Cartesian versus UnWRAPs (Fig 8A). However, UNWRAP provided a narrower error variation for SNR ≥ 30 dB. ECA measured AIF $T_{peak}$ more accurately with UnWRAP trajectories–with a smaller variation–compared to standard

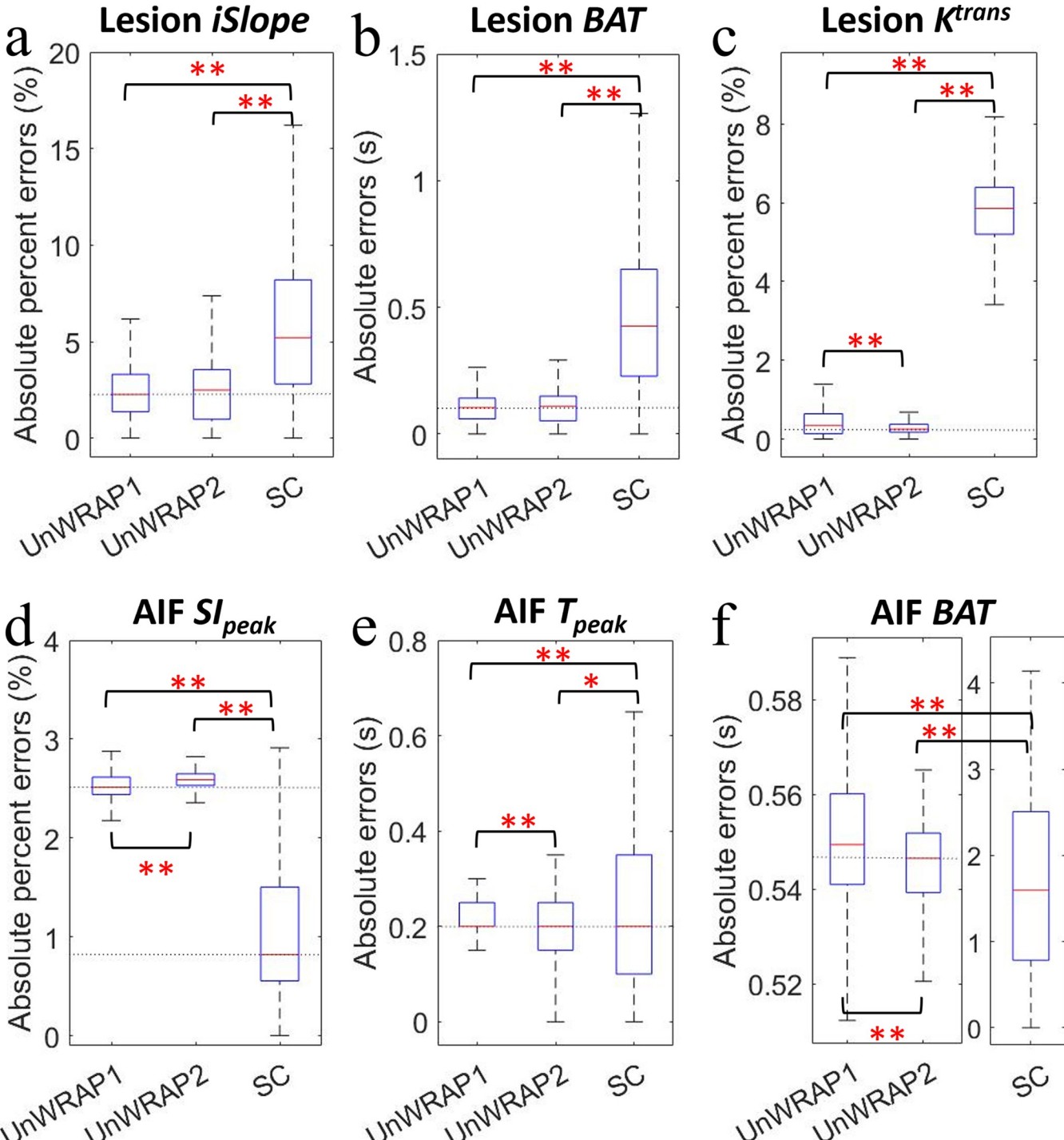

**Fig 6. Absolute percent error or Absolute error in lesion kinetics measured from images by ECA with different sampling trajectories.** ECA acceleration factor = 14. All images were reconstructed without noise. (a) Lesion initial relative enhancement slope (*iSlope*), (b) Lesion *BAT*, (c) Lesion $K^{trans}$, (d) AIF peak signal intensity ($SI_{peak}$), (e) AIF time at peak ($T_{peak}$), and (f) AIF *BAT*. On each box, the central red line indicates the median, and the bottom and top edges of the box indicate the 25th and 75th percentiles, respectively. The whiskers extend to the most extreme data points not considered outliers. SC: Standard Cartesian. Asterisk mark (*) means statistical significance, **: p < 0.0001 and *: p < 0.05.

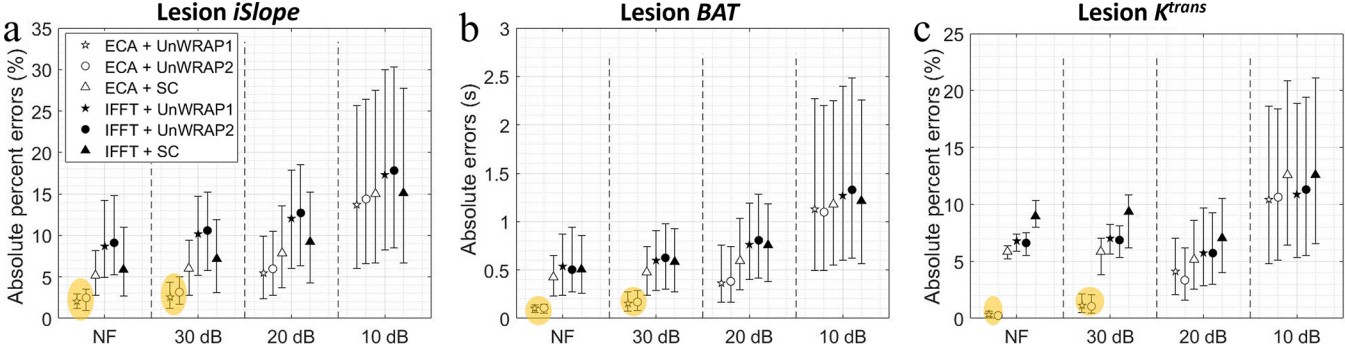

**Fig 7. Absolute percent error or Absolute error in lesion kinetics measured from images by ECA and IFFT with different SNRs. ECA acceleration factor = 14.** NF means noise-free. Makers show median, error bars represent the quarter percentiles. (a) Lesion initial relative enhancement slope (*iSlope*), (b) Lesion *BAT*, and (c) Lesion $K^{trans}$. Yellow ellipse highlighted the best combination of reconstruction method and sampling trajectory for the lowest errors. SC: Standard Cartesian.

Cartesian sampling when SNR ≥ 20 dB. Moreover, ECA reconstruction with UnWRAP trajectories showed superior performance in estimation of AIF *BAT* than standard Cartesian trajectories over all tested noise level. UnWARP trajectories with medium SNR (SNR ≥ 20 dB, equivalent overall noise std ≤ 10%) can ensure minor errors in estimating AIF kinetics, i.e. the median absolute percent error of $SI_{peak}$< 5%, median absolute error of $T_{peak}$ and $BAT$ < 1 s.

## 3.3. ECA performance with different acceleration factors (AF)

Figs 9 and 10 show the absolute (percent) error in lesion and AIF kinetics, respectively, measured from images reconstructed with ECA at different acceleration factors when SNR = 30 dB. The difference between the performance of ECA with different acceleration factors was much smaller than the variation due to noise. The difference was even smaller for kinetic measurements in lesion voxels and for measurements of AIF peak and time of the AIF peak, where the signal changed more slowly and smoothly. However, when measuring parameters that require high-temporal resolution, e.g. AIF *BAT*, the lowest median absolute error was obtained with an acceleration factor of 14 (from the nominal temporal resolution of 7 s to accelerated temporal resolution of 0.5 s).

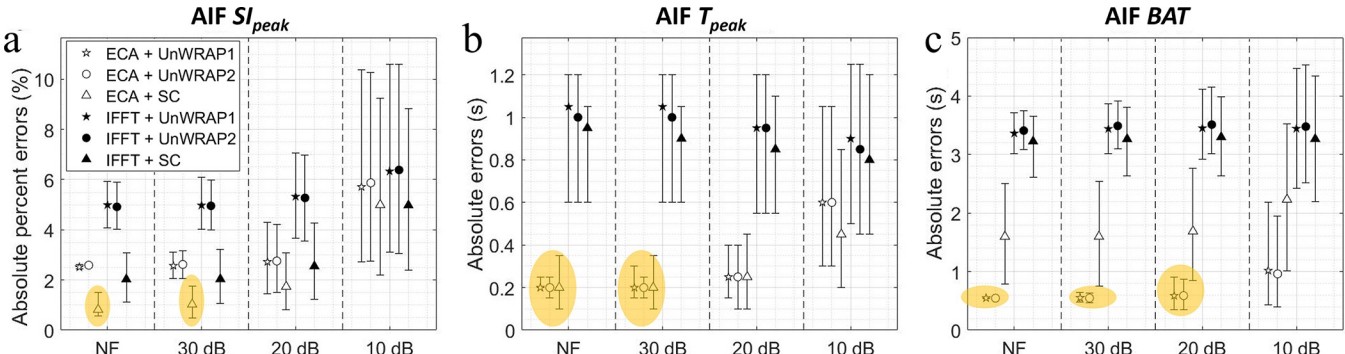

**Fig 8. Median absolute error or median absolute error in AIF kinetics measured from images by ECA and IFFT with different SNRs.** ECA acceleration factor = 14. NF means noise-free. Makers show median, error bars represent the quarter percentiles. (a) AIF peak signal intensity ($SI_{peak}$), (b) AIF time at peak ($T_{peak}$), and (c) AIF *BAT*. Yellow ellipse highlighted the best combination of reconstruction method and sampling trajectory for the lowest errors. SC: Standard Cartesian.

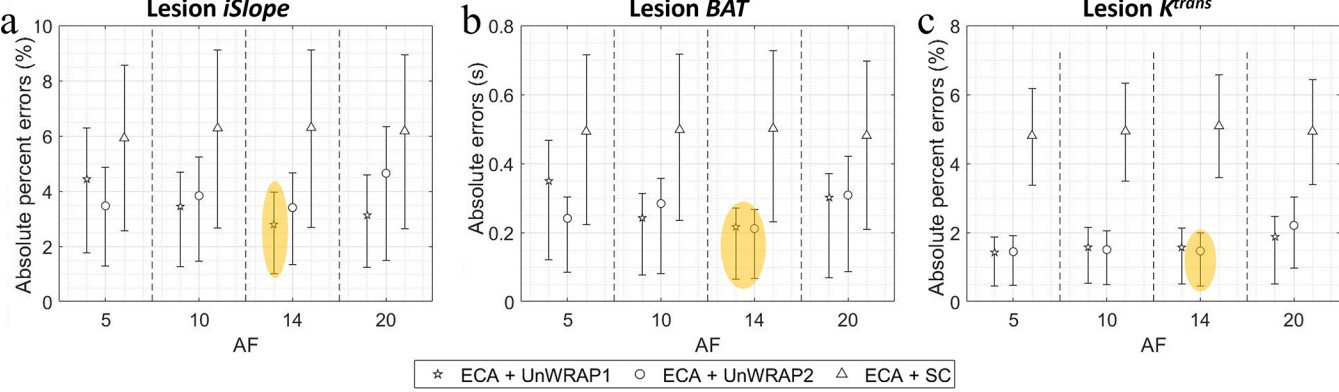

**Fig 9. Median absolute percent error or median absolute error in lesion kinetics measured from images by ECA with different acceleration factor (AF) when SNR = 30 dB.** Makers show median, error bars represent the quarter percentiles. (a) Lesion initial relative enhancement slope ($iSlope$), (b) Lesion $BAT$, and (c) Lesion $K^{trans}$. Yellow ellipse highlighted the best combination of reconstruction method and sampling trajectory for the lowest errors. SC: Standard Cartesian.

### 3.4. Flow phantom experiment

Eighteen ROIs were selected (Fig 11A) representing different locations in the coiled tube. The SNR of the flow phantom image was 22 dB. Fig 11A and 11B show (a) the 2D cross-sectional image of the flow phantom and (b) PSE curves of three ROIs of the flow phantom image reconstructed by 10-fold ECA, standard IFFT with and without Makima interpolation, and ground truth. Fig 11C shows the absolute percent enhancement errors of the flow phantom image reconstructed by standard IFFT (Makima interpolated) and ECA reconstruction.

Table 5 lists the absolute error of the parameters obtained from the measured images compared to the ground truth with 10-fold UnWRAP1 ECA and standard IFFT with (mIFFT) and without Makima interpolation. The absolute errors in measurements of $SI_{peak}$ and $T_D$ were significantly lower by ECA reconstruction than IFFT (p < 0.00001) or mIFFT (p < 0.05). The Makima interpolation significantly increased the accuracy of $T_D$ measurements (p < 0.00001) but did not improve the fidelity of $SI_{peak}$ measurements.

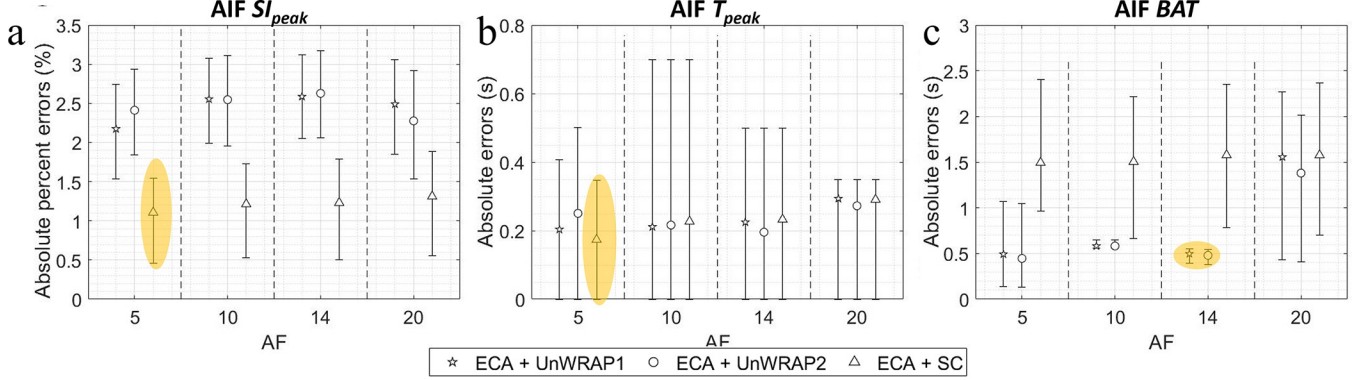

**Fig 10. Median absolute percent error or median absolute error in AIF kinetics measured from images by ECA with different acceleration factor (AF) when SNR = 30 dB.** Makers show median, error bars represent the quarter percentiles. Makers show median, error bars represent the quarter percentiles. (a) AIF peak signal intensity ($SI_{peak}$), (b) AIF time at peak ($T_{peak}$), and (c) AIF $BAT$. Yellow ellipse highlighted the best combination of reconstruction method and sampling trajectory for the lowest errors. SC: Standard Cartesian.

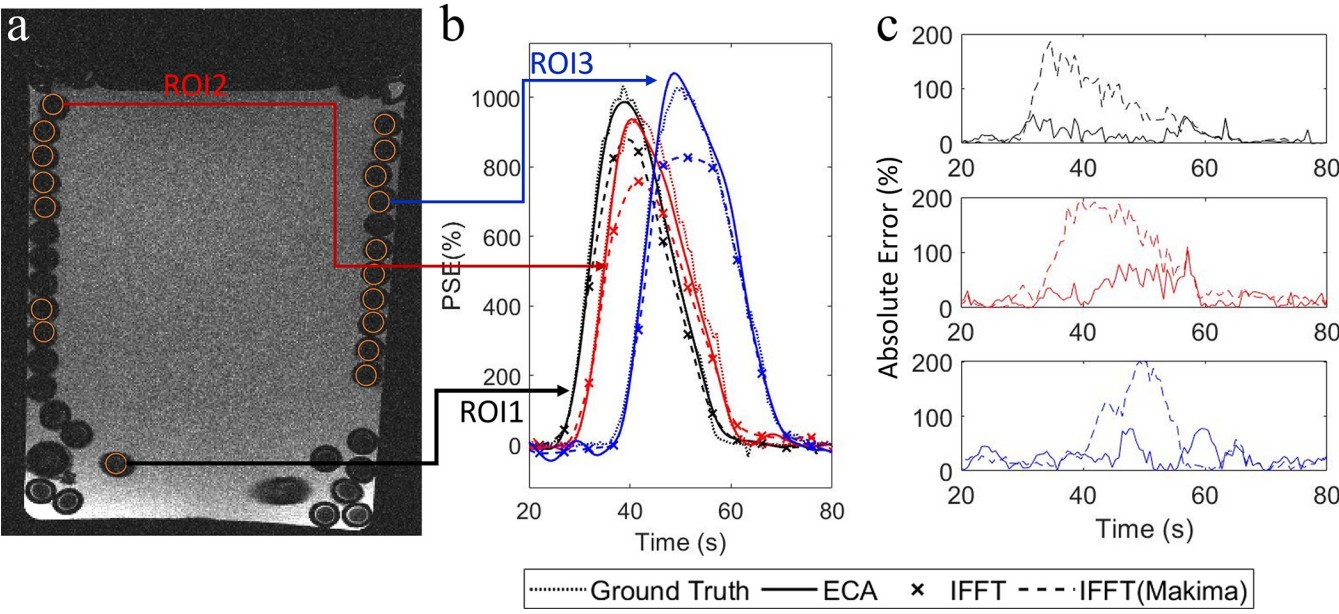

**Fig 11.** Flow phantom experiment (a) 2D cross-sectional image with 18 ROIs representing different locations in the tube, (b) mean PSE curves for three ROIs of flow phantom image reconstructed by 10-fold ECA, standard IFFT with and without Makima interpolation, and ground truth, (3) the absolute mean percent enhancement errors in the three ROIs of the phantom image reconstructed by standard IFFT (Makima interpolated) and 10-fold UnWRAP1 sampled ECA reconstruction.

## 4. Discussion

Both simulation and flow phantom studies demonstrate that the ECA method can reconstruct high-temporal-resolution dynamic images using small partitions of k-space. This allows for more accurate measurements of the kinetics of contrast media distribution through arteries and tissues. ECA accurately estimates kinetic features such as lesion *iSlope* and $K^{trans}$. In addition, ECA provides diagnostic information that is not available from low-temporal resolution imaging, such as *BAT* in tissues/lesions, and accurate recovery of the AIF.

To ensure that each subset of k-space measured during each reconstructed time frame maintains sufficient SNR and morphologic information for accurate ECA image reconstruction, the UnWRAP trajectories are suggested as they uniformly sample spatial frequencies during each time frame. The difference between UnWRAP1 and UnWRAP2 is small compared to the difference between standard Cartesian and UnWRAP trajectories. UnWRAP2 has slightly better performance than UnWRAP1 in terms of higher accuracy when estimating $K^{trans}$, BAT for lesions and AIF, and AIF $T_{peak}$, probably because the UnWRAP2 trajectory samples k-space more uniformly during each time frame than UnWRAP1.

**Table 5. Median value and the 95% confidence interval (CI) of the absolute error of parameters measured from flow phantom images by ECA, IFFT with (mIFFT) and without Makima interpolation (IFFT).**

| Absolute error of Parameters | ECA | | mIFFT | | IFFT | |
|---|---|---|---|---|---|---|
| | Median | 95% CI | Median | 95% CI | Median | 95% CI |
| $SI_{peak}$ | **0.296** | **[0.209, 0.382]** | 1.99 | [1.85, 2.12] | 2.05 | [1.92, 2.18] |
| $T_D$ (s) | **0** | **[-0.042, 0.042]** | 0.317 | [0.273, 0.361] | 6.38 | [5.26,7.51] |

Bold highlights the lowest error of the three methods

$SI_{peak}$: Peak signal intensity

$T_D$: Delay time between the signal enhancement in ROI1 and the other ROIs.

Images reconstructed from UnWRAP k-space sampling trajectories with ECA reconstruction provide more accurate measurements of the AIF and lesion kinetics than the standard IFFT, with a lower absolute error and smaller error variance, particularly with medium to high SNR (SNR≥20 dB). Increasing noise level introduces significant errors in parameters estimated by both ECA and IFFT. High SNR (≥30 dB) is required to ensure minor errors in measurements of lesion kinetics (<5% in median absolute percent error of *iSlope* and $K^{trans}$ or <1 s in median absolute error of *BAT* in lesion). For measuring the AIF, a lower overall SNR (SNR≥20 dB) is sufficient to ensure a minor error (<5% in median absolute percent error of $SI_{peak}$ or <1 s in median absolute error of $T_{peak}$ and *BAT* in vessels). Our previous measurements show that these SNRs are achievable in current routine clinical practice [1].

UnWRAP trajectories improve the performance of ECA reconstruction. However, they do not work well with standard IFFT. IFFT with standard Cartesian k-space sampling can provide smaller absolute (percent) error in estimation of lesion *iSlope* and *BAT*, and all AIF features than IFFT with UnWtRAPs at any SNR.

Different from most regularization methods that are used for compressed sensing with incoherent non-Cartesian sampling [28–31], the ECA reconstruction can be implemented with any sampling trajectory, although the performance will be different depending on the sampling during each reconstruction interval. This means that the ECA method has potential for improved sampling of retrospective data (if the time-stamped k-space raw data were saved). ECA can also be used with current MRI protocol without changing the sampling trajectory. In addition, given the high SNR k-space data, the ECA method can achieve high acceleration with a factor of 14, as validated by this simulation study, which is much higher than the other reported MRI acceleration methods [26, 27].

A large acceleration factor of ECA requires high quality of measurement data. A greater number of under-determined k-space points for each new time frame would reduce SNR and increase reconstruction errors. The sparsity of subtraction images acquired with ultrafast -DCE-MRI during the early phase of enhancement may allow higher acceleration factors with acceptable errors. In this research, we tested four acceleration factors which increased the nominal temporal resolution of 7 s/image to 1.4 s/image—0.35 s/image. The smallest median absolute (percent) error or error variance in pharmacokinetic parameters was achieved with 14-fold acceleration (accelerated temporal resolution with ECA was 0.5 s/image) when SNR = 30 dB. We recognize that our study has some limitations:

- T2* effects were not simulated. This effect is significant immediately after contrast bolus administration, particularly in arteries, and can cause errors in kinetic measurements.

- The effect from artifacts, such as motion or Gibbs ringing, was not simulated in this study.

- Our study only discussed Cartesian-based k-space sampling trajectories. It will be important to validate the effectiveness of ECA reconstruction with non-cartesian trajectories, such as radial [44] or PROPELLER [45].

- We did not compare ECA quantitively with the other reconstruction methods including compressed sensing, view sharing, higher parallel imaging acceleration, etc.

In conclusion, this study demonstrates that ECA reconstruction of k-space data sampled by UnWRAP trajectories can generate high-precision DCE-MR images from short time intervals of k-space data. This allows accurate tracking of contrast bolus propagation and accurate pharmacokinetic measurements, particularly during the critical early phase of enhancement. Accurate measurements of these parameters are critical for increased diagnostic accuracy.

## Acknowledgments

We would like to thank Dr. Timothy J. Carroll in the University of Chicago for the support and advice in flow phantom experiment.

## Author Contributions

**Conceptualization:** Zhen Ren, Ty O. Easley, Federico D. Pineda, Gregory S. Karczmar.

**Data curation:** Zhen Ren, Ty O. Easley, Federico D. Pineda.

**Formal analysis:** Zhen Ren, Ty O. Easley.

**Funding acquisition:** Gregory S. Karczmar.

**Investigation:** Zhen Ren, Ty O. Easley, Federico D. Pineda.

**Methodology:** Zhen Ren, Ty O. Easley, Federico D. Pineda, Rina F. Barber.

**Project administration:** Gregory S. Karczmar.

**Resources:** Gregory S. Karczmar.

**Software:** Zhen Ren, Ty O. Easley, Federico D. Pineda.

**Supervision:** Federico D. Pineda, Rina F. Barber, Gregory S. Karczmar.

**Validation:** Zhen Ren, Xiaodong Guo.

**Visualization:** Zhen Ren.

**Writing – original draft:** Zhen Ren.

**Writing – review & editing:** Zhen Ren, Ty O. Easley, Federico D. Pineda, Xiaodong Guo, Rina F. Barber, Gregory S. Karczmar.

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
