## [Decision Letter · Decision Letter 0]

7 Feb 2023

PONE-D-22-23285Enhancement-Constrained Acceleration (ECA) reconstruction-based high temporal resolution DCE-MRI for pharmacokinetic of breast MRI dataPLOS ONE

Dear Dr. Ren,

Thank you for submitting your manuscript to PLOS ONE. After careful consideration, we feel that it has merit but does not fully meet PLOS ONE’s publication criteria as it currently stands. Therefore, we invite you to submit a revised version of the manuscript that addresses the points raised during the review process.

As pointed by both reviewers, please reformulate the Introduction section and improve the overall quality and style following the reviewers' comments. One weakness of this work is the lack of the comparsion with peer techniques, please add new supporting materials and comment on this point. 

We look forward to receiving your revised manuscript.

Kind regards,

Zhentian Wang, Ph.D.

Academic Editor

PLOS ONE

Journal Requirements:

“This study was supported by a grant from the National Institutes of Health under project number 5R01CA218700-04, which was awarded to G.S.K. The funders had no role in study design, data collection and analysis, decision to publish, or preparation of the manuscript.”

Additional Editor Comments (if provided):

Reviewers' comments:

Reviewer's Responses to Questions

**Comments to the Author**

1. Is the manuscript technically sound, and do the data support the conclusions?

Reviewer #1: Yes

Reviewer #2: Yes

2. Has the statistical analysis been performed appropriately and rigorously? 

Reviewer #1: Yes

Reviewer #2: Yes

3. Have the authors made all data underlying the findings in their manuscript fully available?

Reviewer #1: Yes

Reviewer #2: Yes

4. Is the manuscript presented in an intelligible fashion and written in standard English?

Reviewer #1: Yes

Reviewer #2: Yes

5. Review Comments to the Author

Reviewer #1: I found this paper interesting. Also the presented technique ECA seems promising compared to the conventional methods of accerelating image acquisition. There are some points needing clarification.

#1 Introduction: Interesting, yet a bit length for original paper. It is not a review paper on image reconstruction of UF-DCE MRI and therefore the contents are better to be summarized.

#2 Methodology 2.1 Simulation Phantoms

Table 1 right column"Temporal resolution range 2.8-7

I am just wondering why there are wide variations in temporal resolution.

#3 Considering that non-technically-oriented readers are there, mathematical parts might be better put into the appendix.

#4 Fgure 5 all of the upper row and all of the lower row looks quite similar. Can you magnify parts of the graph to enhance their differece?

Reviewer #2: This is a follow-up study of the performance of the previously proposed temporally constrained reconstruction with different sampling trajectories, SNRs, and acceleration rates in estimating contrast media kinetics in lesions and arteries. The evaluation was performed in simulation and controlled phantom studies. The original publication demonstrated an initial promise of the method, and the current method adds further substantial knowledge on the practical utility of the developments. Overall, the body of knowledge introduced in the manuscript is substantial and may be of interest to PLOS ONE readership. However, there are several moderate-to-minor concerns about the work outlined below:

1. The main weakness of the manuscript is the lack of comparison of the ECA method with peer techniques. Given that the presented results relate to rather artificial situations (simulations and phantom studies), it is important to study if the method actually provides benefits compared to existing state of art. Please pick a related established temporal regularization method and study ECA performance with respect to the benchmark in some relevant experiment(s).

2. Overall quality and style:

Introduction is too long and contains many rationalizations and reviews related to the method which was already published. Instead, it should take into account that the method was already presented and concentrate on the importance of validating the method in controlled conditions. I envision a significant redesign of the Introduction.

Similarly, “Image reconstruction” section contains a very detailed description of the ECA method, which was already described in the supplemental material of the previous paper. I don’t think such detailed duplication is necessary.

The title of the previous paper was “Enhancement constrained acceleration: A robust reconstruction framework in breast DCE-MRI” and the title of the manuscript under review is virtually the same. Instead, it should reflect the essence of the efforts attempted in the manuscript (validation, evaluation, etc).

3. Please cite prior art on the use of the first and second differences in the temporal dimension (e.g., use most recent MRM 2023, 89(1):112-127 to track the relevant references).

6. PLOS authors have the option to publish the peer review history of their article (what does this mean?). If published, this will include your full peer review and any attached files.

Reviewer #1: No

Reviewer #2: No

---

## [Author Response · Author response to Decision Letter 0]

29 Mar 2023

RESPONSE TO DECISION LETTER

Dear Editor and Reviewers,

Thank you so much for your consideration of our paper. We are grateful to receive the valuable comments from reviewers, which are very helpful in improving our paper. We have followed your instructions to modify our paper and would like to submit a revised version. The major changes we have made are (1) shortening the introduction by summarizing the acceleration methods; (2) removing the mathematical description of ECA method; (3) replacing Figure 5; (4) adding introduction and discussion of the regularization methods. We believe that the manuscript has been improved and hope it has reached the standard of the journal. We have addressed each comment as below. In the revised manuscript, we have colored the changes related to Reviewer #1’s comments in green, related to Reviewer #2’s comments in blue, and related both in yellow. 

Reviewer #1: I found this paper interesting. Also the presented technique ECA seems promising compared to the conventional methods of accelerating image acquisition. There are some points needing clarification.

#1 Introduction: Interesting, yet a bit length for original paper. It is not a review paper on image reconstruction of UF-DCE MRI and therefore the contents are better to be summarized.

Authors’ response: Thank you for this suggestion. We have shortened the introduction and summarized acceleration methods. The changes are highlighted in yellow.

#2 Methodology 2.1 Simulation Phantoms

Table 1 right column"Temporal resolution range 2.8-7

I am just wondering why there are wide variations in temporal resolution.

Authors’ response: Thank you for your question. The variations in temporal resolution come from the ultrafast DCE-MRI on different scanners. A higher temporal resolution – approx. 3 seconds per image – was on a 3T MRI scanner. The 7 seconds per image scans were on a 1.5 T scanner.

#3 Considering that non-technically-oriented readers are there, mathematical parts might be better put into the appendix.

Authors’ response: Thank you for your suggestion. We have removed the detailed description of the ECA method and cited a reference that provides a detailed description of ECA reconstruction. The changes are highlighted in yellow.

#4 Figure 5 all of the upper row and all of the lower row looks quite similar. Can you magnify parts of the graph to enhance their difference?

Authors’ response: Thank you for your suggestion. Figure 5 has been changed. We have magnified the part of graph to enhance the difference, but for the two UnWRAP sampling trajectories, the difference is small. The changes are highlighted in green. 

Reviewer #2: This is a follow-up study of the performance of the previously proposed temporally constrained reconstruction with different sampling trajectories, SNRs, and acceleration rates in estimating contrast media kinetics in lesions and arteries. The evaluation was performed in simulation and controlled phantom studies. The original publication demonstrated an initial promise of the method, and the current method adds further substantial knowledge on the practical utility of the developments. Overall, the body of knowledge introduced in the manuscript is substantial and may be of interest to PLOS ONE readership. However, there are several moderate-to-minor concerns about the work outlined below:

1. The main weakness of the manuscript is the lack of comparison of the ECA method with peer techniques. Given that the presented results relate to rather artificial situations (simulations and phantom studies), it is important to study if the method actually provides benefits compared to existing state of art. Please pick a related established temporal regularization method and study ECA performance with respect to the benchmark in some relevant experiment(s).

Authors’ response: We appreciate your questions and comments. We believe that the current ECA method and the application proposed here are very different from existing regularization methods. A useful comparison of the ECA with other regularization methods would require significant modifications and many experiments. Therefore we respectfully suggest that comparison with of the ECA with other regularization method should be left to our next paper. Our goal here is simply to demonstrate that the ECA is effective and has potential to improve clinical MRI. We did not intent to suggest that the ECA is better than other acceleration method. It is likely that different methods may be preferable in different contexts. 

We point out that we DID compare the ECA with interpolation which is one of the simplest regularization methods. In this revision, rather than adding additional simulations, we briefly compared the ECA with other published regularization methods. Most of the existing state-of-the-art methods for image acceleration (which we believe this point refers to) are based on non-Cartesian sampling schemes and in some cases (i.e., compressed sensing approaches) incoherent sampling is required. While we agree that a comparison with other methods would be valuable to benchmark the performance of the proposed method, our purpose in this manuscript and in developing the ECA approach is to introduce a method that does not rely on a specific sampling scheme and is therefore applicable to data acquired under varying conditions. In this manuscript, we opted for testing Cartesian-based sampling trajectories given that these are universally available regardless of vendor or software packages available at different sites. Because the ECA does not require a specific sampling scheme, we do plan to test it under non-Cartesian sampling conditions in future work. At that point hope to benchmark its performance in those experiments against other existing methods. We have added a paragraph in the Introduction and Discussion to further explain the difference between the ECA and other existing regularization methods. We also point out in the discussion that we have not yet tested the ECA for real breast scans and have not compared the ECA with other reconstruction methods. In the revised manuscript, the changes are marked in blue.

2. Overall quality and style:

Introduction is too long and contains many rationalizations and reviews related to the method which was already published. Instead, it should take into account that the method was already presented and concentrate on the importance of validating the method in controlled conditions. I envision a significant redesign of the Introduction.

Authors’ response: Thank you for your suggestion. We have shortened the introduction and summarized the review on the acceleration methods. The changes are highlighted in yellow.

Similarly, “Image reconstruction” section contains a very detailed description of the ECA method, which was already described in the supplemental material of the previous paper. I don’t think such detailed duplication is necessary.

Authors’ response: Thank you. We have removed the introduction of the ECA method and cited a reference of the detailed description of ECA reconstruction. The changes are highlighted in yellow.

The title of the previous paper was “Enhancement constrained acceleration: A robust reconstruction framework in breast DCE-MRI” and the title of the manuscript under review is virtually the same. Instead, it should reflect the essence of the efforts attempted in the manuscript (validation, evaluation, etc).

Authors’ response: Thank you for your suggestion. We have changed the title to “Pharmacokinetic Analysis of Enhancement-Constrained Acceleration (ECA) reconstruction-based high temporal resolution breast DCE-MRI” and highlighted in blue.

3. Please cite prior art on the use of the first and second differences in the temporal dimension (e.g., use most recent MRM 2023, 89(1):112-127 to track the relevant references).

Author’s response: Thank you for this suggestion. We added a paragraph in the Introduction to summarize those works, which is highlighted in blue.

We would like to thank you for all the comments and suggestions. We look forward to the outcome of your assessment.

Yours Sincerely,

Zhen Ren, Ph.D.

Department of Radiology

The University of Chicago

zren1@uchicago.edu

---

## [Decision Letter · Decision Letter 1]

10 May 2023

Pharmacokinetic Analysis of Enhancement-Constrained Acceleration (ECA) reconstruction-based high temporal resolution breast DCE-MRI

PONE-D-22-23285R1

Dear Dr. Ren,

We’re pleased to inform you that your manuscript has been judged scientifically suitable for publication and will be formally accepted for publication once it meets all outstanding technical requirements.

Kind regards,

Zhentian Wang, Ph.D.

Academic Editor

PLOS ONE

Additional Editor Comments (optional):

Reviewers' comments:

Reviewer's Responses to Questions

**Comments to the Author**

1. If the authors have adequately addressed your comments raised in a previous round of review and you feel that this manuscript is now acceptable for publication, you may indicate that here to bypass the “Comments to the Author” section, enter your conflict of interest statement in the “Confidential to Editor” section, and submit your "Accept" recommendation.

Reviewer #1: All comments have been addressed

Reviewer #2: All comments have been addressed

2. Is the manuscript technically sound, and do the data support the conclusions?

Reviewer #1: Yes

Reviewer #2: Yes

3. Has the statistical analysis been performed appropriately and rigorously? 

Reviewer #1: Yes

Reviewer #2: Yes

4. Have the authors made all data underlying the findings in their manuscript fully available?

Reviewer #1: Yes

Reviewer #2: Yes

5. Is the manuscript presented in an intelligible fashion and written in standard English?

Reviewer #1: Yes

Reviewer #2: Yes

6. Review Comments to the Author

Reviewer #1: I think the author responded to my previous queries and the new revised paper becomes easier to follow.

Reviewer #2: The manuscript was significantly improved. Most of my previous comments were addressed except for comparison with existing techniques. While the arguments of not doing such comparison are weak, I think the manuscript is generally in good shape.

7. PLOS authors have the option to publish the peer review history of their article (what does this mean?). If published, this will include your full peer review and any attached files.

Reviewer #1: No

Reviewer #2: No

---

## [Editor Report · Acceptance letter]

7 Jun 2023

PONE-D-22-23285R1 

Pharmacokinetic Analysis of Enhancement-Constrained Acceleration (ECA) reconstruction-based high temporal resolution breast DCE-MRI 

Dear Dr. Ren:

I'm pleased to inform you that your manuscript has been deemed suitable for publication in PLOS ONE. Congratulations! Your manuscript is now with our production department. 

Kind regards, 

on behalf of

Prof. Zhentian Wang 

Academic Editor

PLOS ONE